# ctDNA for the Evaluation and Management of EGFR-Mutant Non-Small Cell Lung Cancer

**DOI:** 10.3390/cancers16050940

**Published:** 2024-02-26

**Authors:** Aakash Desai, Tadana A. Vázquez, Keishla M. Arce, Marcelo Corassa, Philip C. Mack, Jhanelle E. Gray, Bruna Pellini

**Affiliations:** 1Division of Hematology and Oncology, Department of Medicine, University of Alabama at Birmingham, Birmingham, AL 35294, USA; aakashdesai@uabmc.edu; 2School of Medicine, Ponce Health Sciences University, Ponce, PR 00716, USA; tvazquez20@stu.psm.edu (T.A.V.);; 3Thoracic Oncology Unit, BP—A Beneficência Portuguesa de São Paulo, São Paulo 01323-001, Brazil; marcelo.corassa@bp.org.br; 4Center for Thoracic Oncology, The Tisch Cancer Institute, Mount Sinai Health System, New York, NY 10029, USA; 5Department of Thoracic Oncology, H. Lee Moffitt Cancer Center and Research Institute, Tampa, FL 33612, USA; 6Department of Oncologic Sciences, Morsani College of Medicine, University of South Florida, Tampa, FL 33602, USA

**Keywords:** ctDNA, EGFR, MRD, molecular response

## Abstract

**Simple Summary:**

Lung cancer patients with a specific gene mutation often undergo tissue biopsies to guide treatment. But what if a simple blood test could do the same? This review explores “liquid biopsies”, which detect circulating tumor DNA (ctDNA) in the blood, offering several advantages. For early-stage disease, it could help decide on treatment after surgery. In advanced stages, it can identify genetic changes in the tumor that might affect treatment response. Additionally, tracking ctDNA levels can monitor treatment progress and detect resistance early. While tissue biopsies remain standard, liquid biopsies are becoming a recommended alternative. Ongoing research aims to improve test sensitivity for early detection, refine treatment approaches based on ctDNA results, and explore other bodily fluids for even better detection. Making liquid biopsies more accessible and integrated into routine care could significantly improve lung cancer management.

**Abstract:**

Circulating tumor DNA (ctDNA) offers a new paradigm in optimizing treatment strategies for epidermal growth factor receptor (EGFR) mutant non-small cell lung cancer (NSCLC). Its potential spans early-stage disease, influencing adjuvant therapy, to advanced disease, where it aids in identifying genomic markers and resistance mechanisms. This review explores the evolving landscape of utilizing liquid biopsies, specifically circulating tumor DNA (ctDNA), in the management of NSCLC with *EGFR* mutations. While tissue-based genomic testing remains the cornerstone for clinical decision-making, liquid biopsies offer a well-validated, guideline-recommended alternative approach. Ongoing trials integrating ctDNA for *EGFR*-mutant NSCLC management are also discussed, shedding light on the potential of ctDNA in early-stage disease, including its applications in prognostication, risk stratification, and minimal residual disease detection post-curative intent treatment. For advanced disease, the role of ctDNA in identifying resistance mechanisms to EGFR tyrosine kinase inhibitors (TKIs) is explored, providing insights into disease progression and guiding treatment decisions. This review also addresses the challenges, including the limitations in sensitivity of current assays for disease recurrence detection, and calls for future studies to refine treatment approaches, standardize reporting, and explore alternative biofluids for enhanced sensitivity. A systematic approach is crucial to address barriers to ctDNA deployment, ensuring equitable access, and facilitating its integration into routine clinical practice.

## 1. Introduction

Mutations in the epidermal growth factor receptor (EGFR) tyrosine kinase are observed in approximately 15% of NSCLC adenocarcinomas in the United States and occur more frequently in nonsmokers [1]. In Asian populations, the incidence of EGFR mutations is substantially higher, and it is identified in up to 62% of patients [2]. The use of EGFR tyrosine kinase inhibitors (TKIs) is contingent upon the detection of these mutations, which may be detected either in solid tissue biopsies or in liquid biopsies [3]. Traditionally, genomic testing of tumor tissue has been the gold standard for molecular profiling [4]. However, there are considerable innate limitations to tissue testing, including obtaining tumor samples, scarcity of DNA in biopsy samples, and turnaround time for test results [5]. Liquid biopsies have emerged as an attractive alternative for the detection of tumor-derived somatic alterations in a myriad of bodily fluids, with blood being the most commonly used and studied sample [6].

While liquid biopsies encompass the analysis of multiple analytes including, but not limited to, circulating tumor cells, circulating-free RNA, tumor-derived exosomes, the most studied analyte is circulating tumor DNA (ctDNA) [7]. The latter consists of fragments of DNA shed by cancer cells into the plasma of patients with malignancies, which often represents a small fraction of the total circulating cell-free DNA [5,7]. The levels of ctDNA may vary across different cancer types, and higher ctDNA concentrations are associated with larger tumor volume and advanced disease stage [8]. ctDNA can provide a complementary or alternative approach to tissue-based genomic testing and is now increasingly incorporated into clinical practice [4]. Besides the current well-established clinical applications of ctDNA for molecular profiling and the detection of mechanisms of resistance to target therapies, ctDNA is under study for lung cancer screening, molecular response monitoring, minimal residual disease (MRD) detection after curative-intent treatment to enable risk stratification, and further guide treatment decision-making [9].

In this review, we discuss the role of ctDNA and its broad clinical application in the management of patients with early-stage and advanced *EGFR* mutant NSCLC. Furthermore, we discuss challenges, future directions, and ongoing trials, which will provide further clarity on the role of ctDNA and its utility to inform clinical care recommendations.

## 2. ctDNA Analyses vs. Tumor Tissue Genomic Analyses in NSCLC

Multiple guidelines endorse the adoption of a multigene next-generation sequencing (NGS) approach for tumor molecular testing for patients with advanced stage NSCLC [10,11]. As routine comprehensive genomic profiling by NGS has increasingly become standard, there has been a concomitant and dramatic increase in the number of NSCLC patients now eligible for targeted therapies [11,12]. One of the most common driver abnormalities in non-squamous NSCLC is activating mutations in *EGFR*. The identification of *EGFR* driver mutations in advanced NSCLCs has revolutionized treatment, with EGFR TKIs being the standard first-line systemic therapy in this subgroup of patients [13,14]. In parallel, sufficient evidence has evolved for the clinical utility of genotyping advanced NSCLC using ctDNA [4]. ctDNA may be used as an alternative or complementary approach to tumor-biopsy-based molecular testing [4]. Thompson et al. demonstrated that the inclusion of plasma-based NGS testing led to higher rates of guideline-recommended treatment (74% vs. 46%, *p* = 0.0005) compared to traditional tissue NGS [15]. Also, patients who underwent plasma-based NGS had a significantly shorter time before receiving their first-line treatment (12 days vs. 20 days, *p* = 0.0003), particularly those with specific driver mutations (10 days vs. 19 days *p* = 0.0001) [15]. Of note, in the former study, physicians had test results available before the first patient visit much more frequently when a plasma-based strategy was used (85% vs. 9%, *p* < 0.0001) [15]. Moreover, in a retrospective analysis of 170 newly diagnosed NSCLC patients treated at two cancer centers within a 5-year period, liquid biopsy NGS returned results on average 26.8 days faster than tissue and reported higher testing success [16]. For guideline-recommended biomarkers, liquid biopsy results had high rates of tissue concordant results (94.8% to 100%) [16]. When comparing testing modalities, a “liquid-first” approach identified guideline-recommended biomarkers in 76.5% of patients vs. 54.9% in a tissue-first approach [16]. There was no significant difference in time-to-treatment or survival outcomes (overall survival and progression-free survival) based on liquid vs. tissue biopsy findings [16].

With regards to *EGFR*-mutant (*EGFRm*) NSCLC, a meta-analysis of 40 studies estimated a pooled sensitivity of 68% (95% CI = 60–75%) and specificity of 98% (95% CI = 95–99%) for *EGFR* mutation detection in ctDNA [17]. The diagnostic odds ratio was 88 (95% CI = 40–195), with the area under the curve of 0.91 (95% CI = 0.88–0.93). This analysis demonstrated that peripheral blood liquid biopsy had a good specificity for detecting *EGFR* mutation in NSCLC patients, while tissue biopsy still needs to be undertaken for negative blood biopsy patients due to its lower sensitivity [17]. While in patients with *EGFRm* NSCLC, the predominant ctDNA use is for molecular profiling to inform clinical decision-making and for the detection of resistance mechanisms after EGFR TKIs, ctDNA is under study for monitoring molecular response to guide treatment escalation for patients with metastatic *EGFR*-mutant NSCLC (NCT04410796). Table 1 and Table 2 highlight various studies that integrate ctDNA for the management of *EGFR*m NSCLC that are currently underway.

Importantly, liquid biopsy testing may differ based on a priori knowledge of a patient’s tumoral genomic landscape at the time of testing. In a tumor-informed approach (TIA), whole exome sequencing or whole genome sequencing is performed on a patient’s tumoral sample to design a unique panel of mutations that will be interrogated in the plasma samples [18]. This integration helps to identify specific genetic alterations associated with the patient’s tumor, aiding clinicians in tracking disease progression, determining the benefit of treatments, and making personalized treatment decisions [19,20]. A tumor-agnostic approach, on the other hand, also known as tumor-uninformed or tumor-naive, does not require a tissue sample and involves comprehensive mutation calling using a pre-established gene panel [21,22,23,24]. Unlike tumor-informed approaches, uninformed approaches uses an “off-the-shelf” panel, and while it may comparatively have a lower sensitivity to detect genomic alterations, it allows the detection of de novo alterations not previously linked to the underlying disease [25].

In general, liquid biopsy offers a faster turnaround time compared to tissue NGS testing, with a median Turnaround Time (TAT) of approximately 1–2 weeks [26]. When considering other requirements, the typical amount of blood drawn for many applications ranges from 10 to 18 milliliters [27]. Furthermore, it is essential to note that several ‘preanalytical’ factors, such as clotting, DNA leakage from white blood cells (WBCs) and hematopoietic cells, freeze-thawing, DNAse activity in the blood, PCR compatibility of reagents, the time interval between blood drawing and analysis, and temperature during circulating tumor DNA (ctDNA) analysis, may have an impact on the testing results [28].

Previous work from Leighl et al. [29] on molecular profiling for advanced NSCLC using comprehensive ctDNA analysis using Guardant360^®^ CDx demonstrated that the false-positive rates for *EGFR* common mutations were 0% (specificity = 100%) and the false-negative rates ranged from 10 to 18.2% (sensitivity = 81.8–100%) [29]. Published work by Aggarwal et al. [30] found that 6.3% of patients with NSCLC overall had a targetable mutation detected uniquely by plasma ctDNA-based profiling. Further, based on analysis by Gray et al. [31], the expectation is that approximately 15–32% of patients with EGFR-mutant NSCLC do not shed enough ctDNA into circulation to be detected by the current available assays, thus leading to false-negative results. In this scenario, it is important to ensure comprehensive tissue genomic is obtained, which remains the gold standard for metastatic/advanced NSCLC molecular profiling. Finally, some commercial assays such as FoundationOne^®^ Liquid CDx include information on tumor fraction per patient sample in their report. This informs clinicians whether the sample contains enough ctDNA to safely detect genomic alterations. If the tumor fraction is low (<1%), the rate of false negatives is higher requiring tissue testing to confirm whether a genomic alteration is present or not, especially in non-squamous lung carcinoma [32]. Further, if a liquid biopsy result was uninformative, there is no rationale for repeating it immediately after the first liquid biopsy test. However, at the time of disease progression, a repeat liquid biopsy should be considered to interrogate mechanisms of resistance. For patients initiating treatment without taking tissue analysis into consideration, assuming there was insufficient quantity, a repeat tissue biopsy is recommended, especially at the time of disease progression to understand resistance patterns and to assess for histological transformation.

## 3. Role of ctDNA in Early-Stage *EGFRm* NSCLC

ctDNA is emerging as an important tool for prognostication and risk stratification in the early-stage setting. For early-stage, it is important to note that assay sensitivity may depend on tumor size and burden [33]. A study, analyzing TIA ctDNA at different stages of NSCLC, showed diagnostic sensitivities of 64%, 82%, and 100% for tumor stages I, II, and III, respectively [33]. Despite this, as clinical sensitivity of ctDNA-based assays improves, they may prove to be useful as predictive biomarkers after curative-intent therapy such as surgery and radiation [34]. Detection of ctDNA MRD may indicate disease relapse, providing the opportunity for an early intervention and hopefully improving patient’s prognosis.

For patients with completely resected *EGFRm* stage IB to IIIA NSCLC, adjuvant osimertinib for three years is currently approved based on the ADAURA phase III trial, which demonstrated that adjuvant osimertinib improved two-year DFS rates relative to placebo (89% vs. 52%; HR 0.20, 99% CI 0.14–0.30) [35]. These benefits were maintained at a longer follow up of 44 months (DFS HR 0.23) [36], with a subsequent report of improvement in overall survival (OS) (five-year OS rate of 88% vs. 78% with placebo; HR 0.49, 95% CI 0.34–0.70) [37]. Although the drug was well-tolerated, the optimal duration of adjuvant osimertinib is an area of active debate due to concerns with toxicity and cost of care. Prospective ctDNA evaluation can help to identify subsets of patients who require a shorter duration of treatment and ctDNA MRD monitoring can also play a pivotal role in sparing patients with low-risk disease from unnecessary treatments, thus minimizing the potential side effects and healthcare costs.

Serial monitoring of ctDNA in the post-surgical resection setting may help risk stratification, assessment of treatment response, and monitoring for disease relapse [38,39]. The TRACERx study used a tumor-informed approach (anchored-multiplex PCR (AMP)) to detect MRD in early-stage NSCLC after surgery [19,20]. Among patients with recurrence, ctDNA was detected either at the time of relapse or before, with a median lead time of 151 days. In contrast, patients who remained cancer-free were rarely ctDNA-positive [19]. The study also showed that ctDNA could detect relapse earlier than standard imaging methods, proving especially beneficial in minimizing the potential side effects associated with the additional treatment with radiation when opting for serial ctDNA testing [19,40]. Meanwhile, Qiu et al. utilized ultra-deep targeted NGS to evaluate the clinical utility of ctDNA for dynamic recurrence risk and adjuvant chemotherapy benefit prediction in resected NSCLC [41]. The authors demonstrated that ctDNA positivity preceded radiological recurrence by a median of 88 days and that post-surgical and post-chemotherapy ctDNA positivity are significantly associated with worse recurrence-free survival, thus potentially providing a high-risk population where treatment escalation or prolongation might delay recurrence and improve survival [41].

Although ctDNA has great potential as a biomarker for “real-time” reflection of disease burden in response to therapy, the current sensitivity of commercial assays to detect disease recurrence is suboptimal [25]. Also, questions remain regarding the timing of ctDNA sampling after surgery, the number of timepoints to be interrogated to guide treatment escalation or de-escalation, and whether TIA or TAA approaches should be preferred. Several studies sought to answer the question regarding timing of ctDNA MRD interrogation after surgery [19,23,33,42]. The DYNAMIC study investigated peri-operative dynamics of ctDNA in patients with stage I-IIIA NSCLC to determine the appropriate detection time of ctDNA MRD following surgery. It revealed that ctDNA measurements (by tumor-naïve cSMART assay), taken as early as three days post-surgery, can predict patient outcomes, as evidenced by a higher recurrence-free survival in patients with non-detectable ctDNA compared with patients with detectable ctDNA (637 vs. 278 days, *p* = 0.002) [42]. However, others advocate for at least 1–2 weeks after surgery as the ideal timepoint for the first ctDNA MRD interrogation [33]. Jung et al. reported data from 278 patients with AJCC v7 stage I–IIIA EGFR-mutated NSCLC who underwent curative surgery and had ctDNA measured pre-operatively, 4 weeks after curative surgery, and then every 3 months for the first year, every 4 months for the second year, every 6 months for the third year, and thereafter once every year until 5 years or clinically definite recurrence [43]. Patients were then classified into three groups: baseline ctDNA-negative, baseline ctDNA-positive but post-operative MRD-negative, and baseline ctDNA-positive and post-operative MRD-positive. The authors found that the 3-year DFS rate was significantly different among the three groups (84% versus 78% versus 50%, *p* = 0.02), suggesting ctDNA MRD status-based risk stratification and feasibility of longitudinal monitoring of ctDNA to detect early recurrence [43]. However, as this becomes implemented into clinics, it would be important to ensure that longitudinal monitoring and detection of early recurrence is balanced with healthcare costs and financial toxicity as we develop paradigms for evaluation of MRD and the emergence of resistance post-operatively. Furthermore, prior to incorporation into clinical practice, consensus around timing of measurements and assay methodology are critical and should be based on robust clinical trial evidence.

## 4. Role of ctDNA in Advanced *EGFRm* NSCLC

Apart from its established role in comprehensive genomic profiling for advanced NSCLC, ctDNA enables the identification of resistance mechanisms at the time of disease progression [44]. It can also serve as a supplementary disease monitoring strategy by helping with risk stratification during treatment [45,46].

The phase 3 FLAURA trial established osimertinib as the standard first-line therapy for *EGFRm* advanced NSCLC given prolonged PFS compared with standard EGFR-TKIs (18.9 vs. 10.2 months, HR 0.46, CI 95% 0.37–0.57, *p* < 0.0001) [13]. The study also demonstrated the benefit in favor of osimertinib on decreased problems to the central nervous system (CNS) and overall survival (38.6 vs. 31.8 months, HR 0.80, CI 95% 0.64–1.00, *p* 0.0046) [13]. However, patients eventually experience disease progression with osimertinib through various resistance mechanisms [13]. In the era of first-generation TKIs, the T790M mutation was well-described as the major mechanism of resistance. For osimertinib, designed to inhibit T790M-carrying isoforms, there are multiple alternative EGFR-dependent mechanisms of resistance, the most common being the emergence of C797S and multiple rarer mutations, including L792X, G796S, L718Q, S768I, G796R, G796D, and G724S [44]. *EGFR* amplification and copy number alteration are also on-target mechanisms for secondary drug resistance of osimertinib [13,47]. EGFR-independent (off-target) mechanisms of resistance include *MET* amplification (5% to 10% of cases), *HER2* amplification, activation of the RAS–mitogen-activated protein kinase (MAPK) or RAS–phosphatidylinositol 3-kinase (PI3K) pathways, novel fusion events, and histological/phenotypic transformation [47]. Furthermore, genotyping analyses have highlighted differences in the frequency and preponderance of resistance mechanisms when osimertinib is administered in a front-line versus second-line setting, underlying the discrepancies in selection pressure and clonal evolution [48]. These alterations can be detected in tissue genotyping from at-progression biopsies but can also be evaluated on ctDNA, especially when the availability of tissue biopsy is difficult [44]. Considering the potential to detect molecular alterations despite tumor heterogeneity, a liquid-based assay can be of great use in this scenario [49].

Understanding the intricate mechanisms of resistance to targeted agents holds pivotal importance to customize therapeutic interventions aimed at overcoming drug resistance. It is imperative to acknowledge that resistance is inherently an acquired pathological state, mandating the analysis of fresh specimens. In this context, the application of ctDNA emerges as a logical and promising approach for the thorough examination of these resistance mechanisms. In 2013, Murtaza et al. established proof of principle that exome-wide analysis of ctDNA alongside invasive biopsy methods can help identify mutations linked to acquired drug resistance in advanced cancers [50]. The role of ctDNA for advanced oncogene driven NSCLC in identifying resistance mechanisms has continued to evolve. This has already been demonstrated for the emergence of acquired resistance after treatment with first/second-generation *EGFR* TKI’s. Among 116 patients, 11% were found to have resistant-inducing alterations including *MET*, *HER2*, *KRAS,* and *PIK3CA,* indicating that plasma monitoring may enable the rapid identification and early detection of resistant mutations [51].

In the phase 2 APPLE trial [52], the utility of longitudinal plasma *EGFR* T790M monitoring for optimizing the sequencing of gefitinib and osimertinib in patients with advanced *EGFR*-mutant non-small-cell lung cancer was assessed. The trial was designed with three arms: Arm A utilized upfront osimertinib until RECIST-defined progression; Arm B initiated treatment with gefitinib, transitioning to osimertinib upon detection of ctDNA *EGFR* T790M mutation or RECIST progression; and Arm C continued gefitinib until RECIST progression, followed by a switch to osimertinib. The primary endpoint was the 18-month progression-free survival rate ‘on osimertinib’ (PFSR-OSI-18) in Arm B, with a null hypothesis of PFSR-OSI-18 ≤ 40%. In Arm B, 17% (8/47) of patients transitioned to osimertinib due to molecular progression, as indicated by ctDNA T790M emergence, before RECIST-defined progression, with a median time to molecular progression of 266 days. The study successfully met its primary endpoint, demonstrating a PFSR-OSI-18 of 67.2% in Arm B (84% CI: 56.4–75.9%) as compared to 53.5% in Arm C (84% CI: 42.3% to 63.5%), respectively. The median progression-free survival (PFS) was 22.0 months in Arm B versus 20.2 months in Arm C (HR: 0.80, 90% CI 0.51–1.27; *p* = 0.22). The median OS was not reached in Arm B versus 42.8 months in Arm C. The trial substantiates that serial ctDNA T790M monitoring is feasible and facilitates timely transition to osimertinib, resulting in favorable PFS and overall survival outcomes.

There is also an emerging role for ctDNA-based risk stratification, which can further improve patient selection for therapies. Gray et al. reported biomarker analysis from the FLAURA trial, demonstrating the utility of the cobas EGFR mutation tissue and plasma testing [53] to aid selection of patients with *EGFR*m advanced NSCLC for first-line osimertinib treatment [31]. Using reference central cobas tissue test results, positive percent agreements with cobas plasma test results for Ex19del and L858R detection were 79% [95% CI, 74–84] and 68% (95% CI, 61–75), respectively [31]. PFS superiority with osimertinib over comparator EGFR-TKI remained consistent irrespective of randomization route (central/local *EGFR*m-positive tissue test) [31]. Interestingly, in both treatment arms, PFS was prolonged in plasma ctDNA EGFRm-negative (23.5 and 15.0 months) versus positive patients (15.2 and 9.7 months) showing the capability of ctDNA as a robust prognostic marker in this setting [31].

Exploratory analysis from the AURA3 and FLAURA trials showed ctDNA EGFRm analysis as early as 3 weeks on-treatment has the potential to serve as predictive markers of response and may serve as a supplementary monitoring strategy with the potential for early identification and intervention in case of progression [54,55]. Mack et al. conducted analyses of serial plasma ctDNA level measurement at baseline, 8 weeks, and at progression in patients with *EGFR*m NSCLC enrolled in the SWOG S1403 clinical trial of afatinib ± cetuximab [56]. Complete clearance of *EGFR* mutations in ctDNA by 8 weeks was significantly associated with improved PFS and OS compared with those with persistent ctDNA at Cycle 3, Day 1 [57]. Clearance of ctDNA was also associated with a decreased risk of death, suggesting that ctDNA monitoring could be a valuable tool for guiding treatment decisions and improving patient outcomes [57]. Currently, a US-based multi-institutional study is assessing whether treatment escalation with the addition of platinum-doublet to osimertinib in patients with stage IV *EGFRm* NSCLC who have persistently detectable *EGFR* mutations through ctDNA analysis after 3 weeks of osimertinib will result in an improved duration of response to first-line therapy (NCT04410796).

ctDNA also allows the identification of co-mutations associated with worse prognosis, and its use in this setting may enable personalized treatment selection based on disease risk in a timely manner. For example, co-mutations such as *TP53* or other tumor suppressor genes (*RB1*, *NF1*, *ARID1A*, *BRCA1*, and *PTEN*) are potentially associated with worse prognosis in patients with *EGFRm* NSCLC [58]. Hence, ctDNA-based assessment prior to therapy selection may enable identification of these alterations associated with poor outcomes and define the need for a more aggressive treatment paradigm.

Although osimertinib is currently the standard of care treatment for advanced *EGFRm* NSCLC, recent efforts have focused on improving outcomes for this patient population. Although previous studies such as IMPRESS [59] and NEJ009 [60] demonstrated benefit with combination chemotherapy and tyrosine kinase inhibition, this has not generally been incorporated in practice given that the studies did not utilize a third-generation TKI such as osimertinib. This year, several studies showed promising results with upfront combination therapies for patients with *EGFRm* NSCLC. FLAURA2 studied the addition of platinum-based chemotherapy to osimertinib as a first-line treatment and demonstrated an improvement in PFS from 16.7 months with osimertinib alone to 25.5 months with osimertinib plus chemotherapy (HR: 0.62, *p* < 0.0001) [61]. Meanwhile, MARIPOSA studied with the addition of amivantamab to third-generation EGFR TKI lazertinib for first-line treatment of locally advanced or metastatic *EGFRm* NSCLC [62]. Compared to osimertinib there was improvement in PFS (23.7 vs. 16.6 months, HR: 0.70 (95% CI 0.58–0.85, *p* < 0.001), including in patients with a history of brain metastases. Lastly, the phase 2 RAMOS trial evaluated patients with metastatic *EGFRm* NSCLC who were treated with osimertinib plus ramucirumab 10 mg/kg via intravenous infusion every 3 weeks compared to osimertinib alone and showed an improved PFS with combination therapy (24.8 vs. 15.6 months (HR, 0.55; 95% CI, 0.32–0.93; *p* = 0.023) [63]. Although these approaches demonstrate improved PFS, improvement in OS remains to be seen and these regimens will need to be carefully balanced with the increased toxicity and costs associated with these combination therapies. In this regard, ctDNA-based kinetics and co-mutation analyses may serve as risk-stratifying tools to select patients for a more aggressive treatment approach consisting of one of the described combination treatment strategies in lieu of osimertinib alone in the first-line setting.

## 5. Challenges and Future Directions

In the context of thoracic malignancies, ctDNA testing has emerged as a valuable tool in advanced stages; however, considerable gaps persist in understanding its role for MRD detection in early-stage settings. The current commercially available MRD assay exhibits limited and variable clinical sensitivity, impeding its application in clinical decision-making [25]. To overcome this, various approaches including increasing the number of tracked mutations or adding DNA methylation to the current assays, parallel WBC-sequencing, use of epigenetic DNA modifications, and physical DNA fragmentation patterns are currently being studied [64,65]. Moreover, robust clinical trial evidence is imperative to elucidate its prognostic and risk stratification capabilities, thereby influencing clinical recommendations and decision-making processes [9]. In the advanced stage, further evidence is required to comprehend the potential implications of ctDNA monitoring and whether early treatment interventions could alter the disease’s natural progression, particularly in the context of *EGFR*-mutated NSCLC. Although these applications show promise, their routine clinical integration necessitates additional evidence.

Other challenges associated with ctDNA implementation in clinical settings include the lack of standardized ctDNA tests reporting and false-positive results due to the detection of germline variants and clonal hematopoiesis of indeterminate potential (CHIP) [5]. From a technical standpoint, clonal hematopoiesis introduces biological noise in ctDNA analysis, impacting the test’s specificity [66]. As such, as these tests become part of treatment selection, escalation or de-escalation, structured referral mechanisms for genetic consultation, and hematologic evaluation will be essential for discussing and managing results revealing germline mutations or clonal hematopoiesis of indeterminate potential (CHIP). Further, as the field evolves, improved clinician education on interpretation of ctDNA results will be imperative to avoid treatment changes based on false-positive tests.

The distinctive advantage of liquid biopsy lies in its versatility across various stages of cancer care as a diagnostic (for mutation detections), predictive and prognostic tool. While current use-cases predominantly involve ctDNA assays, these represent only the initial applications, with emerging roles in MRD detection and cancer screening. A systematic approach is imperative to address potential barriers to the effective deployment of ctDNA technologies, encompassing economic burden, financial toxicity, logistical constraints, and disparities in testing accessibility.

Future studies should concentrate on refining treatments for resectable and advanced *EGFR*-mutated NSCLC (Figure 1). In resectable cases, ctDNA MRD holds potential for informing the duration of adjuvant osimertinib, serving as a prognostic biomarker for early relapse detection and, eventually, for treatment escalation as a method to improve cure rates. Meanwhile, in locally advanced or metastatic diseases, ctDNA not only guides the accurate identification of resistance mechanisms but also has the potential to improve disease risk stratification. This can identify patients for whom treatment escalation or de-escalation may be beneficial, simultaneously reducing treatment-related costs and toxicity.

## 6. Conclusions

In conclusion, the analysis of circulating tumor DNA (ctDNA) in EGFR-mutant non-small cell lung cancer plays a critical role in various facets of patient management. It is instrumental in diagnosis, prognostication, identification of resistance mechanisms, and, increasingly, in monitoring treatment response and guiding therapeutic decisions. Furthermore, the emerging potential of ctDNA analysis in informing treatment escalation and de-escalation strategies offers promising avenues for enhancing treatment efficacy and reducing adverse effects.

## Figures and Tables

**Figure 1 cancers-16-00940-f001:**
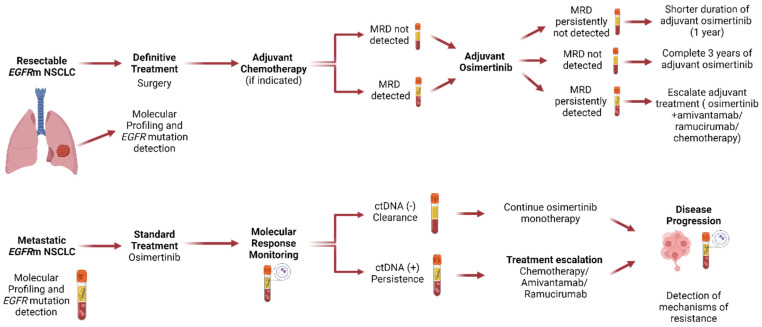
Proposed clinical trial design for *EGFR*-mutant non-small cell lung cancer using ctDNA to personalize treatment decision-making. Patients with resectable disease treated with surgery +/− adjuvant chemotherapy should have MRD status assessed by ctDNA after curative-intent treatment and start adjuvant osimertinib. Patients who remain ctDNA MRD (−) should de-escalate therapy and be treated for 1 year with osimertinib; patients who become ctDNA MRD (−) should complete 3 years of adjuvant osimertinib; patients who have persistently ctDNA MRD (+) should escalate therapy to add either chemotherapy, amivantamab, or ramucirumab. Patients with metastatic disease receive first-line osimertinib. Patients who become ctDNA (−)/clearance should continue osimertinib monotherapy; patients that persistently have ctDNA (+) should undergo treatment escalation with the addition of either chemotherapy, amivantamab, or ramucirumab. ctDNA, circulating tumor DNA; MRD, minimal residual disease.

**Table 1 cancers-16-00940-t001:** Clinical trials for advanced EGFR-mutant non-small cell lung cancer using ctDNA as an evaluation or intervention.

Trial (NCT #)	Intervention	Study Design	Primary Outcome	Study Population	N	Recruiting Status
**NCT04841811 (APPROACH)**	Almonertinib (Continuous versusMRD guided)	Phase III	ORR EFS	Unresectable, stage III with *EGFR* mutations	192	Recruiting
**NCT06020989 (CHAMELEON)**	Lazertinib followed by added Pemetrexed + Carboplatin if ctDNA-positive versus Lazertinib monotherapy	Phase II Parallel Assignment Randomized	PFS	Metastatic, first line, sensitizing *EGFR* mutations (Ex19del/L858R) in patients without ctDNA clearance after induction Lazertinib	129	Not yet recruiting
**NCT04912687 (CIRCULAR)**	ctDNA evaluation paired with tissue evaluation for EGFR mutations	Prospective Cohort	EGFR detection rate by combined tissue and ctDNA analisys	Newly diagnosed, advanced, without previous molecular evaluation	580	Recruiting
**NCT05334277 (FOCUS-C)**	Furmonertinib followed by added Carboplatin + Pemetrexed +/− Bevacizumab if ctDNA-positive versus Furmonetinib monotherapy	Phase II Parallel Assignment Randomized	PFS	Untreated advanced or metastatic with *EGFR* mutations (del19/ L858R) with or without ctDNA clearance after induction Furmonetinib	280	Recruiting
**NCT03865511 (MELROSE)**	Osimertinib Tissue biopsy and ctDNA analysis	Phase II	Evaluation of genetic profile at disease progression on first-line Osimertinib	Locally advanced or metastatic, common *EGFR* mutations (L858R/Ex19del)	66	Recruiting
**NCT04737382 (OSIRIS)**	Paired tissue NGS and ctDNA analysis	Prospective cohort, interventional	EGFR TKI resistance analysis on tissue biopsies and ctDNA	Metastatic, sensitizing *EGFR* mutations, disease progression on Osimertinib	200	Recruiting
**NCT05281406 (PACE-LUNG)**	Osimertinib followed by added Platinum + Pemetrexed if ctDNA-positive	Phase II	PFS1 (PFS from start of chemotherapy)	Stage IIIB/IV, common *EGFR* mutations (L858REx19del), ctDNA positive at week 3 of Osimertinib	50	Recruiting
**NCT05598528 (PRECISE)**	Almonertinib, Furmonetinib or Osimertinibe Genomic profile evaluation both in tissue and blood (ctDNA)	Prospective cohort	ORR/PFS Differences in genomic profiles (tissue and ctDNA), ctDNA dynamics	Stage IIIB-IV *EGFR* mutant receiving third-generation EGFR-TKIs in first line.	210	Recruiting
**NCT05020275 (RESISTYR)**	ctDNA analysis and Osimertinib Pharmacokinetics	Prospective Cohort Case-Only	PFS related to plasma exposure to Osimertinib.	Advanced / metastatic with *EGFR* mutations on first-line Osimertinib	60	Recruiting
**NCT05257967**	ctDNA evaluation on CSF paired with ctDNA in blood	Prospective Cohort	Concordance of ctDNA on CSF and plasma	*EGFR* mutations, leptomeningeal disease	10	Recruiting
**NCT05401110**	Cohort II: Osimertinib + Carotuximab	Phase I Single arm	Dose limiting toxicities	Advanced or metastatic, *EGFR* mutations, with ctDNA positivity after 12 weeks of first-line Osimertinib	60	Recruiting
**NCT05534113**	Almonertinib followed by Envafolimab (after ctDNA clearance)	Phase II Single Arm	PFS TRP	Unresectable, advanced, with *EGFR* mutations and PD-L1-positive	38	Not yet recruiting
**NCT05536505**	Icotinib or Osimertinib if MRD detected post-surgery versus observation if MRD not detected.	Phase II Parallel Assignment Non-Randomized	DFS 3y-DFS	Resectable, Stage IB-IIIB, *EGFR* mutations, MRD evaluation post-surgery.	180	Recruiting
**NCT05813522**	Furmonertinib + CSF ctDNA clearance analysis	Phase II	PFSi PFSo	*EGFR* mutations, leptomeningeal disease	30	Enrolling by invitation

**Abbreviations:** 3y-DFS (3-year Disease-Free Survival), CSF (Cerebrospinal Fluid), ctDNA (Circulating Tumor DNA), EGFR (Epidermal Growth Factor Receptor), EFS (Event-Free Survival), MRD (Minimal Residual Disease), NGS (Next-Generation Sequencing), ORR (Objective Response Rate), PD-L1 (Programmed Death Ligand 1), PFS (Progression-Free Survival), PFSo (Overall Progression-Free Survival), PFSi (Intracranial Progression-Free Survival), TKI (Tyrosine Kinase Inhibitor), TRP (Transient Receptor Potential). Trial status was evaluated at clinicaltrials.gov, last update in December 2023.

**Table 2 cancers-16-00940-t002:** Clinical trials for early-stage EGFR-mutant non-small cell lung cancer using ctDNA as an evaluation or intervention.

Trial (NCT #)	Intervention	Study Design	Primary Outcome	Study Population	N	Recruiting Status
**NCT04712877 (LCMC LEADER/LCMC4)**	ctDNA analysis Tumor NGS	Prospective cohort	Proportion of patients with actionable oncogenic drivers	Early stage (IA2-III), potentially resectable and operable.	1000	Recruiting
**NCT05079022**	ctDNA followed by adjuvant furmonetinib for 3 years	Phase II	Clearance of ctDNA in 6 months.	Stage I, resected, *EGFR* mutations with ctDNA-MRD positivity post-surgery.	50	Not yet recruiting
**NCT06053099 (ROSIE)**	ctDNA analysis (multiple timepoints) FFPE blocks (surgical specimen)	Prospective cohort	Feasibility (Patients receiving Osimertinib after 12 months of enrollment)	Completely resected, stage IB-IIIA, with common *EGFR* Mutations (L858R/Ex19del).	300	Not yet recruiting

**Abbreviations:** ctDNA (Circulating Tumor DNA), NGS (Next-Generation Sequencing), EGFR (Epidermal Growth Factor Receptor), MRD (Minimal Residual Disease), FFPE (Formalin Fixed Paraffin Embedded). Trial status was evaluated at clinicaltrials.gov, last update in December 2023.

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
