# Peer review of "ctDNA for the Evaluation and Management of EGFR-Mutant Non-Small Cell Lung Cancer"

_cancers, 2024, doi:10.3390/cancers16050940_

Round 1
Reviewer 1 Report
Comments and Suggestions for Authors
In this review article, the authors have provided an overview and current advance of ctDNA in EGFR-mutant non-small cell lung cancer (NSCLC) treatment. This review is well written and would be an important source in understanding the use of ctDNA in clinical practice. While ongoing clinical studies of ctDNA testing in NSCLC management have been summarized in Table 1 and Table 2, the completed clinical trial results can also be summarized in the tables. The article can be further improved by including tables or figures that summarize the key findings, as well as the challenges, in this field.
Reviewer 2 Report
Comments and Suggestions for Authors
The authors have adeptly navigated the dynamic landscape of employing liquid biopsies, particularly circulating tumor DNA (ctDNA), in NSCLC management with EGFR mutations. The review serves as a valuable resource, and its impact and accessibility could be further strengthened by addressing the following points.
Major Comments:
1. It is recommended to explicitly highlight the primary objectives in the introduction for better clarity.
2. While the review extensively covers potential ctDNA applications across disease stages, summarizing key findings concisely and emphasizing their clinical implications, including ctDNA types, liquid biopsy detection times, and peripheral blood testing volumes for each discovery, would enhance clarity.
3. Addressing challenges, such as limitations in sensitivity for disease recurrence detection, is crucial. Provide specific details on ongoing efforts or potential solutions to overcome these limitations.
4. The discussion on ongoing trials integrating ctDNA for EGFR-mutant NSCLC management is insightful, shedding light on the current research landscape. To deepen the analysis, consider offering a more detailed examination of the methodologies, patient cohorts, and interim findings of these trials. Additionally, discussing how the outcomes might influence or reshape current clinical practices would elucidate the potential impact of ctDNA integration. This nuanced exploration would provide readers with a comprehensive understanding of the clinical relevance and implications of these trials in EGFR-mutant NSCLC management.
Comments on the Quality of English LanguageMinor editing of English language required
Reviewer 3 Report
Comments and Suggestions for Authors
Indeed, circulating tumor DNA (ctDNA) offers a new paradigm for optimizing treatment strategies for epidermal growth factor receptor (EGFR)-mutated non-small cell lung cancer. Its potential spans both early stage disease, influencing adjuvant therapy, and late stage disease, where it helps identify genomic markers and mechanisms of resistance. In this review, the authors examine the evolving use of liquid biopsies, particularly circulating tumor DNA (ctDNA), in the treatment of EGFR-mutated NSCLC.
The review is well written and logically and coherently organizes clinical trial evidence for both early and advanced lung cancer. However, for me a number of questions remain unanswered.
1. When detecting circulating tumor DNA, what is the likelihood of false-positive and false-negative responses? How to check the correctness: by response to treatment? Or is tissue biopsy still the next step and cannot be ruled out?
2. Are there invalid samples and the need for a repeat blood test? If we start treatment without taking tissue for analysis, will a biopsy be possible later or will the results be distorted during treatment?
3. When performing a tissue biopsy, what is the frequency of obtaining unreliable results or an indeterminate answer? Comparable to detection of mutations in circulating tumor DNA?
4. In the tables, data on clinical trials are presented only for those recruiting, are there no completed trials yet? I would like to take into account data from completed clinical trials.
Round 2
Reviewer 1 Report
Comments and Suggestions for Authors
The manuscript has improved and can now be accepted.
Reviewer 2 Report
Comments and Suggestions for Authors
The authors have made the point-by-point responses to the reviewers' comments. I have no more comments. And it can be accepted in present form.
Comments on the Quality of English LanguageNo more comments.
Reviewer 3 Report
Comments and Suggestions for Authors
I have no further comments on the manuscript.